# TtCel7A: A Native Thermophilic Bifunctional Cellulose/Xylanase Exogluclanase from the Thermophilic Biomass-Degrading Fungus *Thielavia terrestris* Co3Bag1, and Its Application in Enzymatic Hydrolysis of Agroindustrial Derivatives

**DOI:** 10.3390/jof9020152

**Published:** 2023-01-23

**Authors:** Azucena López-López, Alejandro Santiago-Hernández, Maribel Cayetano-Cruz, Yolanda García-Huante, Jorge E. Campos, Ismael Bustos-Jaimes, Rodolfo Marsch-Moreno, Claudia Cano-Ramírez, Claudia G. Benitez-Cardoza, María Eugenia Hidalgo-Lara

**Affiliations:** 1Departamento de Biotecnología y Bioingeniería, CINVESTAV-IPN, Av. Instituto Politécnico Nacional No. 2508, Gustavo A. Madero, Ciudad de México CP 07360, Mexico; 2Laboratorio de Bioquímica Molecular UBIPRO, FES Iztacala, UNAM, Av. De los Barrios No. 1, Los Reyes Iztacala, Tlalnepantla, Estado de México CP 54090, Mexico; 3Departamento de Bioquímica, Facultad de Medicina, Universidad Nacional Autónoma de México (UNAM), Coyoacan, Ciudad de México CP 04510, Mexico; 4Laboratorio de Variación Biológica y Evolución, Departamento de Zoología, Escuela, Nacional de Ciencias Biológicas, Instituto Politécnico Nacional, Prolongación de Carpio y Plan de Ayala s/n, Delegación Miguel Hidalgo, Ciudad de México CP 11340, Mexico; 5Laboratorio de Investigación Bioquímica, ENMH-Instituto Politécnico Nacional, Guillermo Massieu Helguera No. 239 La Escalera Ticomán, Ciudad de México CP 07320, Mexico

**Keywords:** *Thielavia terrestris*, cellulase/xylanase, bifunctional enzyme, exoglucanase, purification, thermostability

## Abstract

The biomass-degrading thermophilic ascomycete fungus *Thielavia terrestris* Co3Bag1 produces TtCel7A, a native bifunctional cellulase/xylanase GH7 family. The purified TtCel7A, with an estimated molecular weight of 71 kDa, was biochemically characterized. TtCel7A displayed an optimal pH of 5.5 for both activities and an optimal temperature of 60 and 50 °C for cellulolytic and xylanolytic activities, respectively. The half-lives determined for cellulase activity were 140, 106, and 41 min at 50, 60, and 70 °C, respectively, whereas the half-lives observed for xylanase activity were 24, 10, and 1.4 h at 50, 60, and 70 °C, respectively. The *K*_M_ and V_max_ values were 3.12 mg/mL and 50 U/mg for cellulase activity and 0.17 mg/mL and 42.75 U/mg for xylanase activity. Circular dichroism analysis suggests changes in the secondary structure of TtCel7A in the presence of CMC as the substrate, whereas no modifications were observed with beechwood xylan. TtCel7A displayed the excellent capability to hydrolyze CMC, beechwood xylan, and complex substrates such as oat bran, wheat bran, and sugarcane bagasse, with glucose and cellobiose being the main products released; also, slightly less endo cellulase and xylanase activities were observed. Thus, suggesting TtCel7A has an exo- and endomode of action. Based on the characteristics of the enzyme, it might be considered a good candidate for industrial applications.

## 1. Introduction

Lignocellulosic wastes, such as wheat bran, sugarcane bagasse, crushed citrus fruit peel, hay, corn cob, and municipal solid waste, can be converted into commercially essential products, such as ethanol and glucose [1,2]. Lignocellulosic biomass contains cellulose (20–50%), hemicellulose (15–35%), and lignin (10–30%) on a dry weight basis, with variations depending on the plant species; the cellulose and hemicellulose are the primary plant cell wall polymers. They are polysaccharides linked with β-1,4-glycosidic bonds [3,4,5,6,7].

Degradation of lignocellulosic biomass requires the action of multiple carbohydrate-active enzymes, which act synergistically, complementing their action modes, such as cellulases for cellulose degradation and xylanases for the degradation of xylan, the main fraction of hemicellulose [8,9,10,11,12]. Cellulases generally include three classes of glycoside hydrolases: cellobiohydrolase (EC 3.2.1.91), endo-1,4-β-glucanase (EC 3.2.1.4), and β-glucosidase (EC 3.2.1.21). The action of these enzymes generates products such as oligosaccharides, cellobiose, and glucose [11,13,14,15]. Xylan is the main component of hemicellulose, composed of a linear backbone of β-D-1,4-xylopyranose units, and it is hydrolyzed by two enzymes, endo xylanase (EC 3.2.1.8) and β-xylosidase (EC 3.2.1.37). The action of xylanases generates xylooligosaccharides and monosaccharides [3,7,14,16].

Fungal, bacterial, and archaeal microorganisms can produce lignocellulose degrading enzymes with different applications, including biofuel, food, feed, detergent, textile, pulp, and paper industries because the characteristics of these processes, enzymes that are stable at elevated temperatures are required [9,12,17,18]. The production of xylanases by thermophilic bacteria, such as *Caldicoprobacter algeriensis*, *Acidothermus cellulolyticus*, and *Thermosaccharolyticum*, and some thermophilic xylanases by fungi, such as *Myceliophthora thermophila*, *Humicola insolens*, and *Thermoascus aurantiacus* has been reported [19]. Additionally, several bacteria producing thermophilic cellulases, such as *Bacillus licheniformis*, *Geobacillus stearothermophilus*, *Thermotoga neapolitana*, *Caldicellulosiruptor saccharolyticus*, and *Caldocellum saccharolyticum*, have been described [20].

There are a variety of thermophilic fungi, such as *Acremonium thermophilum*, *Chaetomium thermophilum*, *Humicola grisea*, *Humicola grisea* var. *thermoidea*, *Humicola insolens*, *Melanocarpus albomyces*, *Myceliophthora thermophila*, *Talaromyces emersonii*, and *Thermoascus aurantiacus*, and cellulases produced by these fungi have been studied [13,18,21,22]. Thermophilic fungi grow at temperatures between 20 and 50 °C or above, with optimal growth temperatures between 44 and 55 °C [23,24]. In cellulolytic filamentous fungi, the main secreted enzymes are from the GH family 7 (GH7), which are essential for industrial processes in biomass degradation [4,6,25]. Due to the complexity of lignocellulosic material, a combination of glycosyl hydrolases is necessary for its enzymatic hydrolysis [2]. Bifunctional cellulases–xylanases have been found in fungal enzymes belonging to the GH7 family. These enzymes are considered good candidates for industrial applications, such as the recombinant MtEG7a from Myceliophthora thermophila, ThCel7B from *Trichoderma harzianum*, CTendo7 from *Chaetomium thermophilum*, and CtCel7 from *Chaetomium thermophilum* [26,27,28,29]. 

The thermophilic ascomycete fungus *T. terrestris* is of great interest as it breaks down lignocellulosic biomass and is a source of thermostable enzymes [30]. According to the optimal growth temperature, *T. terrestris* Co3Bag1 has been termed a thermophilic fungus capable of growing at 45 °C or above [30]. Some studies of native glycosyl hydrolases from *T. terrestris* Co3Bag1, such as the exo-β-1,3-glucanase *TtBgnA* [31] and the xylanase *TtXynA* [32], have been reported by our research group; also, a recombinant cellobiohydrolase *CBH7B* from *T. terrestris* was studied [33]. However, to our knowledge, no studies are related to a native GH7 bifunctional cellulase–xylanase from members of the genus *Thielavia*. This work presents the purification and characterization of a 71 kDa native bifunctional cellulase/xylanase GH7 family from the thermophilic fungus *T. terrestris* Co3Bag1, named TtCel7A. Additionally, we describe the application of the enzyme in the enzymatic hydrolysis of agroindustrial derivatives, such as oat bran, wheat bran, and sugarcane bagasse.

## 2. Materials and Methods

### 2.1. Chemicals

Culture media were purchased from BD Difco (Sparks, MD, USA), JT Baker (Phillipsburg, NJ, USA), and Sigma-Aldrich (St. Louis, MO, USA). The chemicals used for the sodium dodecyl sulfate-polyacrylamide gel electrophoresis (SDS-PAGE) analysis and the molecular weight standards (MW) were purchased from Bio-Rad (Hercules, CA, USA). Other chemicals, solvents, and substrates used were of analytical grade and were purchased from Sigma-Aldrich.

### 2.2. Microorganisms and Growth Conditions

The strain *Thielavia terrestris* Co3Bag1 was previously described [32]. For spore production, *T. terrestris* Co3Bag1 was cultured at 45 °C on a solid medium, as defined by Tien and Kirk [34]. For enzyme production, 125 mL Erlenmeyer flasks with 50 mL of basal medium as described by Zouari-Mechichi [35] with the following composition: peptone, 5 g/L; yeast extract, 1 g/L; ammonium tartrate, 2 g/L; KH_2_PO_4_, 1 g/L; MgSO_4_·7 H_2_O, 0.5 g/L; KCl, 0.5 g/L; and 1 mL of trace elements solution were used, and the solution pH was adjusted to 5.5 by the addition of 1 M HCl. Then, Erlenmeyer flasks containing 50 mL of basal medium were supplemented with 1% (*w*/*v*) of CMC as a carbon source, and each flask was inoculated with 5 × 10^6^ spores/mL [34]. Culture flasks were incubated at 45 °C for 5 days and 120 rpm unless otherwise stated.

### 2.3. Cellulase and Xylanase Activity Assays

The cellulolytic activity was measured using the 3,5-dinitrosalicylic acid (DNS) colorimetric method [36] with glucose as standard. The assay mixture contained 20 µL of enzymatic preparation and 980 µL of 50 mM acetate buffer, pH 5.5, containing 0.6% (*w*/*v*) of CMC as substrate; it was then incubated at 60 °C for 10 min. All activity assays were performed in triplicate. One unit of enzyme activity was defined as the amount of enzyme required to release 1 µmol of glucose in 1 min under assay conditions. 

Xylanase activity was measured using the 3,5-dinitrosalicylic acid (DNS) colorimetric method [36] using xylose as standard. The assay mixture contained 20 µL of enzymatic preparation and 980 µL of 50 mM acetate buffer, pH 5.5, containing 0.3% (*w*/*v*) of beechwood xylan as substrate; it was then incubated at 50 °C for 15 min. All the activity assays were performed in triplicate. One unit of enzyme activity was defined as the amount of enzyme required to release 1 µmol of xylose in 1 min under assay conditions. 

### 2.4. Enzyme Purification

The culture supernatant (crude extract) was recovered after 5 days of incubation and adjusted at 80% saturation with ammonium sulfate for protein precipitation. The precipitate was collected by centrifugation (7000 rpm, 4 °C for 15 min) and dialyzed against 50 mM Tris-HCl buffer, pH 7.5, supplemented with 25 mM KCl, 0.1 mM PSFM, and 5% (*v*/*v*) glycerol. The dialyzed protein was loaded onto an anion exchange chromatography column (UNOsphere Q, Bio-Rad, USA). Adsorbed proteins were eluted from the column with a linear gradient of KCl (0.025–1 M) in 50 mM Tris-HCl buffer, pH 7.5, supplemented with 25 mM KCl, 0.1 mM PSFM, and 5% (*v*/*v*) glycerol, at a constant flow rate of 2 mL/min, and 2 mL fractions were collected. Fractions with cellulolytic and xylanolytic activity were pooled and concentrated using centrifuge filters, Amicon^®^ Ultra-15, with a 50 kDa cutoff (Merck Millipore, Billerica, MA, USA). Concentrated protein was loaded onto a gel filtration column (Bio-Gel P-100) at a constant flow rate of 0.20 mL/min, and 2 mL fractions were collected (Appendix A). Fractions of purified enzyme were assayed for enzymatic activity and protein concentration, and an aliquot was loaded in a 10% SDS-PAGE. The purified enzyme was stored at 4 °C for further studies.

### 2.5. Protein Concentration

Protein concentration was measured by the Lowry method [37] using bovine serum albumin (Life Technologies, Grand Island, NY, USA) as standard.

### 2.6. SDS-PAGE and Zymogram Analysis

Proteins were analyzed by 10% SDS-PAGE, according to the method described by Laemmli [38]. Gels were stained with Coomassie Brilliant Blue R-250 (Bio-Rad). Molecular weight (MW) was estimated by linear regression using a broad range molecular weight protein standard (Bio-Rad). Zymogram analysis was carried out as previously described [39] with some modifications. Briefly, protein samples were separated in 10% polyacrylamide gel copolymerized with 1% CMC or 1% Remazol Brilliant Blue linked to xylan (RBB-xylan), which were used as the substrates under denaturing conditions. Protein samples were resuspended in sample buffer without 2-β-mercaptoethanol (2-ME) and then boiled in a water bath for 5 min. After electrophoresis, gels were incubated in 50 mM acetate buffer, pH 5.5, at 47 °C for 30 and 120 min for cellulolytic and xylanolytic activity, respectively. For cellulolytic activity, gels were stained with Congo red (1 mg/mL) for 15 min and destained with 1 M NaCl for 10 min. In contrast, xylanase activity was directly observed for the appearance of a hydrolysis zone.

### 2.7. Carbohydrate Content 

The carbohydrate content of TtCel7A was determined by glycoprotein staining. The protein sample was fractionated by SDS-PAGE under reducing conditions on 10% gel. The qualitative detection was carried on with the periodic acid-Schiff method [40], and the same fraction was stained with Coomassie Brilliant Blue R-250 to visualize the protein band. 

### 2.8. Biochemical Properties of TtCel7A

#### 2.8.1. Optimal pH and pH Stability

The effect of pH on the enzyme activity was evaluated in the pH range of 3–8, using 50 mM buffers at different pH values: citrate (pH 3–6), acetate (pH 4–5.5), citrate-phosphate (pH 3–7), and phosphate (pH 5.5–8). For cellulase activity, the reaction mixture containing an aliquot of enzyme preparation and 0.6% (*w*/*v*) CMC in the appropriate buffer was incubated at 60 °C for 10 min. For xylanase activity, the reaction mixture containing an aliquot of enzyme preparation and 0.3% (*w*/*v*) beechwood xylan in the appropriate buffer was incubated at 50 °C during 15 min.

The enzyme preparation was preincubated for pH stability assay in 50 mM citrate-phosphate buffer at different pH values in the range of 3 to 7 without substrate. Reaction mixtures were incubated at 25 °C for 60 and 30 min for cellulase and xylanase activity, respectively. Subsequently, the remaining cellulolytic or xylanolytic activity was measured under standard conditions.

#### 2.8.2. Optimal Temperature and Thermal Stability Assay

The optimal temperature was determined by measuring the cellulolytic and xylanolytic activity of TtCel7A in 50 mM acetate buffer, pH 5.5, containing 0.6% CMC and 0.3% beechwood xylan, respectively. Reaction mixtures were incubated over 30 to 90 °C for 10 and 15 min for cellulolytic and xylanolytic activities, respectively.

To evaluate thermal stability, TtCel7A was incubated at 50, 60, and 70 °C without substrate in 50 mM acetate buffer, pH 5.5. Aliquots of the sample were withdrawn at different time intervals, and the residual activity was measured under standard conditions. The half-life (t_1/2_), corresponding to 50% of the original activity, was determined under standard conditions and compared with the activity without incubation.

#### 2.8.3. Substrate Specificity and Kinetic Parameters

The effect of the substrate in the enzymatic activity of TtCel7A was determined under optimal assay conditions, using 0.6% (*w*/*v*): CMC, Avicel, beechwood xylan, or laminarin. The kinetic parameters *K*_M_ and V_max_ of TtCel7A were determined under optimal conditions for enzyme activity using CMC and beechwood xylan as substrate at concentrations ranging from 0.2 to 12 mg/mL. The *K*_M_ and V_max_ values were calculated using the Marquardt method for performing nonlinear regression applied to Michaelis and Menten model using GraphPad Prism 8.4.0. The turnover number (k_cat_) for each substrate was calculated using data obtained from the enzymatic kinetic parameters.

#### 2.8.4. Circular Dichroism

Far-UV CD spectra were measured using a JASCO J-815 spectropolarimeter (Jasco Inc., Easton, MD, USA) equipped with a PFD-425S Peltier-type cell holder for temperature control and magnetic stirring. Scans were taken between 200 and 250 nm at a scan rate of 10 nm min-1 using a 1.0 cm path-length cuvette.

#### 2.8.5. Effect of Ions, EDTA, and 2-mercaptoethanol (2-ME) on Enzyme Activity

The effect of metal ions and other compounds on the enzymatic activity of TtCel7A was studied. The purified enzyme was incubated in 0.6% CMC and 0.3% beechwood xylan prepared in 50 mM acetate buffer, pH 5.5, containing CaCl_2_, CoCl_2_, CuCl_2_, FeCl_2_, HgCl_2_, KCl, LiCl, MgCl_2_, MnCl_2_, NaCl, NiCl_2_, EDTA, or 2-ME, at a final concentration of 1 mM under optimal assay conditions (60 °C, 10 min for cellulase, or 50 °C, 15 min for xylanase activity). The 100% of activity was expressed as the activity observed in the absence of any compound.

#### 2.8.6. Analysis of TtCel7A Hydrolysis Products

The hydrolysis products released by the action of TtCel7A on CMC and beechwood xylan as substrates were analyzed using thin-layer chromatography (TLC). The purified enzyme was mixed with 50 µL of 1% (*w*/*v*) of CMC or 1% (*w*/*v*) of beechwood xylan in 50 mM acetate buffer, pH 5.5. Reaction mixtures were incubated at 40 °C, and aliquot samples were taken after 0, 12, 24, 48, and 72 h of incubation. TLC was carried out at room temperature using a mixture of butanol:ethanol:water 5:3:2 (*v*/*v*) as the mobile phase. A total of 3 µL of each sample and 2 µL of cellulose oligomer (glucose, cellobiose, or cellotetrahose) and xylan oligomers (xylose, xylobiose, xylotetraose, or xylohexaose) were spotted on silica gel and used as standard. The plate was sprayed with sulfuric acid (15% *v*/*v*) and then heated in a dry oven at 80 °C for 40 min for color development.

Assays were also performed using complex substrates such as wheat bran, oat bran, and sugarcane bagasse. Substrates were incubated with purified TtCel7A at 40 °C at a final 30 mg/mL concentration. Aliquot samples were withdrawn after 0, 24, 48, and 72 h of incubation, and the production of reducing sugars at different times was measured according to the DNS method. TLC was carried out at room temperature using a mixture of butanol:ethanol:water 5:3:2 (*v*/*v*) as the mobile phase. A total of 3 µL of each sample and 2 µL of cellulose, cellobiose, and xylose were spotted on silica gel and used as standard.

### 2.9. Partial Amino Acid Sequencing of TtCel7A 

Purified TtCel7A protein was sequenced by tandem mass spectrometry (MS/MS) at the Proteomics Unit of the Instituto Nacional de Medicina Genómica (INMEGEN, Ciudad de México, México) and that of the Instituto de Biotecnología (UNAM, Ciudad de México, México). The amino acid sequence of peptides obtained was analyzed using the Paragon algorithm in ProteinPilot™ software (Version 4.5), the UniProt Knowledgebase database, and other tools available at NCBI (https://www.ncbi.nlm.nih.gov/ (accessed on 15 May 2017)) and Expasy (https://www.expasy.org/ (accessed on 15 May 2017)). The sequence of the glycosyl hydrolase GH7 from *T. terrestris* NRRL 8126 (accession number XP_003653508.1) was also used as a query to retrieve sequences from the NCBI database and for primer design from selected highly conserved amino acid regions.

### 2.10. Total RNA Extraction and Synthesis of cDNA from T. terrestris Co3Bag1

A total amount of 1 × 10^6^ spores from *T. terrestris* Co3Bag1 were cultured in 50 mL Zuoari-Mechichi media supplemented with 1% CMC as carbon source for 48 h, at 45 °C and 120 rpm, in a shaker incubator. Then, the culture was centrifuged at 8500 rpm, 4 °C, for 20 min. Supernatant was discharged and mycelia were washed with isotonic solution (NaCl 0.9% *w*/*v*) at the centrifugation conditions previously described. Then, liquid nitrogen was poured directly onto the mycelia, which was scraped until a fine powder was obtained. Aliquots of 100 mg pulverized mycelia were stored at −80 °C until use. An aliquot of pulverized mycelia (100 mg) was utilized for extracting total RNA with the RNeasy Plant mini kit from Qiagen^®^, according to the supplier directions. Total RNA extraction samples were analyzed by gel electrophoresis using 1% agarose dissolved in 0.5X TBE buffer. To amplify the ORF Ttcel7A gene, cDNA was synthesized from the *T. terrestris* Co3Bag1 total RNA by using the In-Fusion^®^ SMARTerTM kit from Clontech^®^. The cDNA obtained was employed as template in a PCR reaction using the Oligo (dT) for the first chain, and SmarterV forward primer from SMARTerTM kit (Clontech^®^) and the reverse primer 5’-GAGGCACTGGTAGTACCAGTC-3′, designed from the 3′end coding sequence of the glycoside hydrolase family 7 from *T. terrestris* NRRL 8126 (accession number G2R5G6) (Appendix A). The PCR program used was as follows: denaturation temperature 95 °C, 3 min, 34 cycles of 95 °C, 30 s; alignment temperature at 57 °C, 30 s; extension temperature 72 °C, 4 min. A final extension temperature of 72 °C for 15 min. The PFU high fidelity DNA polymerase from Thermo Scientific was used for DNA amplification. The PCR product was purified from gel and cloned into the pJET1.2/blunt from Life-Technologies^®^.

### 2.11. Analysis of TtCel7A Encoding Gene

Once ORF Ttcel7A gene was amplified by PCR, it was sequenced by Sanger dideoxy sequencing, and the sequence was deposited in GenBank under the accession number ID: KX426377.1. The nucleotide sequence of the TtCel7A gene from *T. terrestris* Co3Bag1 was analyzed and compared to sequence databases using available online tools at Expasy (https://www.expasy.org (accessed on 18 December 2017)) and at NCBI (https://www.ncbi.nlm.nih.gov (accessed on 18 December 2017)). The theoretical isoelectric point and molecular weight were calculated at Expasy (https://www.expasy.org (accessed on 18 December 2017)) using the online ProtParam tool. The signal peptide was predicted using SignalP-5.0 Server. The NetNGlyc 1.0 Server and NetOGlyc 4.0 Server were used for the glycosylation analysis.

### 2.12. Analysis of the 2D and 3D Structures of TtCelA

The secondary structure of TtCel7A from *T. terrestris* Co3Bag1 was analyzed by using the ESPript program [41]. The three-dimensional (3D) structure of TtCel7A from *T. terrestris* Co3Bag1 was generated by submitting the amino acid sequence to the Phyre^2^ server (http://www.sbg.bio.ic.ac.uk/phyre2/ (accessed on 03 December 2020)) and visualized with PyMOL (https://pymol.org/2/ (accessed on 09 January 2021)).

### 2.13. Phylogenetic Reconstruction

Phylogenetic analysis of bifunctional cellulase/xylanase TtCel7A from *T. terrestris* Co3Bag1 was carried out with endoglucanase of GH7 from *Calycina marina* (KAG9246729.1), *Humicola insolens* (sp|P56680|GUN1_HUMIN), *Myceliopthora thermophila* (sp|G2QCS4|CEL7A_MYCTT), *Coniochaeta* sp. (tr|A0A5N5NYX8|A0A5N5NYX8_9PEZI), *M. thermophila* (tr|G2Q665|G2Q665_MYCTT), and *Thermothielavioides terrestris* (XP_003653508.1); beta-glucosidases of GH3 from *Trichoderma parareesei* (OTA05998.1 and OTA04756.1); and endoglucanase of GH5 from *Aspergillus niger* (GJP88372.1), *Purpureocillium takamizusanense* (XP_047838369.1), *T. simmonsii* (QYT00999.1), and *Cordyceps militaris* (ATY58494.1). Sequences downloaded from Genbank (www.ncbi.nlm.nih.gov/genbank/ (accessed on 15 July 2022)) and Uniprot (https://www.uniprot.org (accessed on 22 July 2022)) were aligned with TtCel7A from *T. terrestris* Co3Bag1 using default parameters in SeaView v. 4.0 [42]. A phylogenetic inference analysis by maximum likelihood was performed with PhyML 3.0 at the ATGC Montpellier Bioinformatics platform (www.atgc-montpellier.fr/phyml (accessed on 22 July 2022)). The best amino acid substitution model for this set of data was estimated with the SMS software [43] in the same platform using the Akaike information criterion (AIC) and was Blosum62+G+F (-InL 14,779.55, gamma parameter 8.079). The robustness of the nodes was assessed after 1000 pseudoreplicates, and the putative endoglucanase sequence of *A. nidulans* (AAT90341.1) of the GH17 family was included as an outgroup.

## 3. Results

### 3.1. Production of Cellulase and Xylanase Activities

The production of cellulase and xylanase activities was evaluated in the culture supernatant of *T. terrestris* Co3Bag1 grown on the basal medium Zouari-Mechichi, supplemented with 1% (*w*/*v*) CMC as carbon source, at 45 °C and 120 rpm. Greatest cellulase activity (3.26 U/mL) was registered after 5 days of incubation. In contrast, two peaks of xylanolytic activity were observed after 5 (0.64 U/mL) and 10 (0.92 U/mL) days of incubation, respectively (Figure 1). Cellulase activity decreased after 7 days of incubation, whereas xylanolytic activity showed almost the same values from 5 to 10 days after incubation.

### 3.2. Enzyme Purification and Zymogram Analysis

A crude extract from the cell-free culture supernatant of *T. terrestris* Co3Bag1 grown on CMC was recovered after 5 days of incubation and used as a source of cellulase and xylanase activities. A summary for the purification stages of TtCel7A is shown in Table 1. TtCel7A was purified with a 63.71-fold purification factor, a specific activity of cellulase (33.13 U/mg) and xylanase (1.96 U/mg), respectively, and a recovery yield of 6.15%. A single band of purified TtCel7A, with an estimated molecular weight of 71 kDa, was observed by SDS-PAGE analysis (Figure 2a), which also displayed both cellulase and xylanase activity with zymogram analysis (Figure 2b,c).

### 3.3. Carbohydrate Content 

The glycosylated nature of TtCel7A was assessed, firstly by staining a semipurified fraction of the enzyme with Coomassie Brilliant Blue R-250 (Figure 3a) and then comparing it after staining with periodic acid-Schiff reagent (Figure 3b), where the 71 kDa band was stained in both cases, implying the glycosylation of the enzyme.

### 3.4. Biochemical Properties of TtCel7A

According to the zymogram analysis, the cellulase TtCel7A displays cellulase and xylanase activities. For this reason, the biochemical characterization of the purified enzyme was carried out for both cellulase and xylanase activities.

#### 3.4.1. Optimal pH and pH Stability

TtCel7A displayed optimal activity for both cellulase and xylanase activities at pH 5.5. Although both enzymatic activities were high for a wide range of pH values from 4 to 7, TtCel7A displayed more than 70% of cellulase activity in the pH range from 4 to 7, whereas the enzyme exhibited more than 80% of xylanase activity in the pH range from 3.5 to 7 (Figure 4a).

The pH stability of cellulase and xylanase activities of TtCel7A was determined in the pH range from 3.0 to 7.0. TtCel7A retained more than 70% of cellulase activity in the pH range from 4.5 to 6.5 after 60 min of incubation at 25 °C. In contrast, the enzyme conserved more than 80% of xylanase activity in the pH range from 5 to 6, after 30 min of incubation at the same temperature (Figure 4b). Nonxylanase activity was detected after 60 min of incubation of TtCel7A at 25 °C, in the pH range from 3.0 to 7.0.

#### 3.4.2. Optimal Temperature and Thermal Stability Assays

TtCel7A manifested optimal temperatures of 60 and 50 °C for cellulase and xylanase activities at pH 5.5, respectively (Figure 5a). However, the enzyme retained more than 60% of cellulase activity in the temperature range from 50 to 80 °C, and more than 75% of the xylanase activity in the temperature range from 40 to 80 °C (Figure 5a).

The thermal stability of the cellulase and xylanase activities of TtCel7A was measured after incubation at 50, 60, and 70 °C, at pH 5.5. The half-lives determined for cellulase activity were 140, 106, and 41 min at 50, 60, and 70 °C, respectively (Figure 5b). In contrast, the half-lives observed for xylanase activity were 24, 10, and 1.4 h at 50, 60, and 70 °C, respectively (Figure 5c).

#### 3.4.3. Substrate Specificity and Kinetic Parameters

The substrate specificity of TtCel7A against CMC, Avicel, beechwood xylan, and laminarin was determined under optimal assay conditions, using 0.6% (*w*/*v*) for each substrate. TtCel7A was fully active on CMC (2.74 U/mL) and was able to hydrolyze beechwood xylan (0.41 U/mL), which corresponds to 15% relative activity compared to that exhibited when CMC is the substrate; however, TtCel7A was not active on either Avicel or laminarin (Appendix A).

The kinetic parameters K_M_ and V_max_ of TtCel7A for cellulase and xylanase activities were determined under optimal conditions, using CMC and beechwood xylan as substrates, at a concentration ranging from 0.2 to 12 mg/mL. The enzyme showed a Michaelis and Menten kinetic for cellulase and xylanase activities with K_M_ and V_max_ values of 3.12 mg/mL and 50 U/mg for cellulase activity. In contrast, K_M_ and V_max_ for xylanase activity were 0.17 mg/mL and 42.75 U/mg, respectively. The k_cat_ values obtained for cellulase and xylanase activity were 0.07 and 3.9 × 10^−3^(s^−1^), respectively (Table 2).

#### 3.4.4. Circular Dichroism

The effect of the substrate in the secondary structure of TtCel7A, in the presence of CMC or beechwood xylan as the substrate was evaluated by circular dichroism. Far-UV CD spectra of TtCel7A, CMC, TtCel7A+CMC mixture, beechwood xylan, or TtCel7A+ beechwood xylan were independently determined. The differential spectrum for CMC (TtCel7A+CMC mixture—CMC) suggests some changes in the secondary structure of TtCel7A, in the presence of CMC as the substrate (Figure 6a). In contrast, the differential spectrum for beechwood xylan (TtCel7A+ beechwood xylan mixture–beechwood xylan) indicates there was no major secondary structure reorganization for TtCel7A in the presence of beechwood xylan as the substrate (Figure 6b).

#### 3.4.5. Effect of Metal Ions, EDTA, and 2-mercaptoethanol (2-ME) on Enzyme Activity

The effect of metal ions, EDTA, and 2-ME on cellulase and xylanase activities of TtCel7A, at a final concentration of 1 mM, was evaluated (Figure 7). A 2.2-fold increase was observed in the cellulase activity of TtCel7A in the presence of the metal ion Mn^2+^; also, there was a 1.8- to 1.6-fold increase for this enzymatic activity with other metal ions such as Co^2+^, Fe^2+^, and Ca^2+^, whereas nonactivity occurred with metal ion Hg^2+^. For xylanase activity, a 1.4-fold increase was observed in the presence of Fe^2+^. In contrast, metal ions such as Mg^2+^, Hg^2+^, and the chelating agent EDTA decreased xylanase activity by 60, 46, and 34%, respectively, compared to a negative control with no additions (Figure 7).

#### 3.4.6. Analysis of TtCel7A Hydrolysis Products

The mode of action of TtCel7A towards CMC, beechwood xylan, oat bran, wheat bran, and sugarcane bagasse were analyzed by thin-layer chromatography (TLC) (Figure 8 and Figure 9). Sugarcane bagasse has a high content of xylan (20.3 ± 0.5%) and polymeric glucose (6.8 ± 0.2%) [44]. After 72 h of incubation with 1% of CMC, it appeared that the main product was glucose (Figure 8a). After 72 h of incubation with 1% of beechwood xylan, the main product was xylobiose (Figure 8b). The TtCel7A enzyme was also notable for its capacity to release reduced sugars from oat bran, wheat bran, and sugarcane bagasse. The main hydrolysis product observed after 72 h of incubation with oat bran and wheat bran was glucose (Figure 9a,b), and the amount of released sugar detected was 3.7 and 7.0 g/L, respectively (Appendix A). With sugarcane bagasse, after 72 h of incubation, the main products released consisted of glucose and cellobiose (Figure 9c), along with 4.2 g/L of sugar (Appendix A).

### 3.5. Analysis of TtCel7A Encoding Gene

The TtCel7A encoding gene (Ttcel7A) (Sequence ID: KX426377.1) from *T. terrestris* Co3Bag1 has an open reading frame (ORF) of 1584 bp, encoding for a predicted protein of 527 amino acid residues. The theoretical pI (isoelectric point) and molecular weight calculated were 4.89 and 54.9 kDa, respectively. The predicted signal peptide has 17 amino acid residues (Figure 10). Bioinformatic analysis of the predicted TtCel7A enzyme showed two potential sites for N-glycosylation, and 39 potential sites for O-glycosylation, and three catalytical residues: Glu230, Asp232, and Glu235 (Figure 10). The green box shows glycosyl hydrolase family 7 (GH7) conserved domain (CD) (positions 22-451 aa) and the blue box indicates the cellulose-binding domain (CBM) (positions 494-527 aa) (Figure 10). Similarity sequence analysis of TtCel7A revealed that the enzyme has a 97% shared identity with the glycoside hydrolase family 7 protein (GH7) from *Thermothielavioides terrestris* NRRL 8126 (XP_003653508.1); 77–75% with glycoside hydrolases family 7 from *Thermothelomyces thermophilus* ATCC 42464 (Access number XP_003660789.1), *Achaetomium luteum* (Access number AGV05131.1), *Canariomyces microsporus* (Access number AGV05127.1), *Corynascus sepedonium* (Access number AGV05125.1), and *Collariella gracilis* (Access number AGV05132.1); and 71–69% with the glycoside hydrolases family 7 from *Chaetomium thermophilum* var. *thermophilum* DSM 1495 (Access number XP_006693547.1), and *Scedosporium apiospermum* (Access number XP_016645836.1).

### 3.6. Identity of TtCel7A by Partial Amino Acid Sequencing

After partial amino acid sequencing of purified TtCel7A, the following peptides were obtained (Appendix A): QQACTLTAENHPTLSWSKCTSGGSCTSVSG, VTIDANWR, AGAKYGTGYCDSQCPR, NGEANNVGWTPSSNDK, YAGECDPDGCDFNSYR, KFTVVTQFLTDSSGNLSEIKRFYVQNGVVIPNSNSNIAGVSGNSITQAFCDAQKTAFGDTNVFDQK, GTCPTTSGVPADVESQAPNSKVIYSNIR, and LNDWYYQCL. The peptides obtained were compared to sequence databases and showed 83% identity with the glycoside hydrolase family 7 from *Thermothielavioides terrestris* NRRL 8126; 73–71% identity with cellulase 1,4-beta-cellobiosidases from *Coniochaeta* sp. 272.1 (KAB5572580.1), *Coniochaeta ligniaria* NRRL 30616 (OIW31667.1), and *Acremonium thermophilum* (CAM98445.1); and 65% identity with an exoglucanase 1 from *Neopestalotiopsis* sp. 37M (KAF3010495.1).

### 3.7. Analysis of 2D and 3D Structure of TtCel7A

Crystal structure data for thermostable cellobiohydrolase Cel7A from the fungus *Humicola grisea* var. *thermoidea* (PDB accession code 4CSI) was used as a template to predict the secondary structure elements of TtCel7A from *T. terrestris* Co3Bag1 and four other cellulases. This analysis allowed to predict 7 α-helices and 30 β-sheets for the five amino acid sequences analyzed, as shown in Figure 10.

The three-dimensional (3D) structure of TtCel7A was generated with the Phyre^2^ server. The best model was built using template PDB 4ZZP from Dictyostelium purpureum cellobiohydrolase Cel7A apo structure, 431 residues were aligned with 100% confidence. The CD was identified with red, the CBM with orange, and the linker region with blue (Figure 11).

### 3.8. Phylogenetic Reconstruction

The bifunctional cellulase–xylanase TtCel7A from *T. terrestris* Co3Bag1 grouped with glycosyl hydrolases from the GH7 family from *T. terrestris* NRRL 8126, *C. marina*, *H. insolens*, *M. thermophila*, and *Coniochaeta* sp., and this cluster was well supported (>99). Glycosyl hydrolases from GH3 and GH5 families were grouped in another cluster with a supported value of 100% (Figure 12).

## 4. Discussion

This paper described the purification and biochemical characterization of GH family 7 bifunctional cellulase/xylanse, named TtCel7A, from the thermophile fungus *T. terrestris* Co3Bag1.

TtCel7A was purified from the culture supernatant of *T. terrestris* Co3Bag1 grown at 45 °C on 1% CMC as the only carbon source. The purified enzyme is a glycoprotein with an estimated molecular weight of 71 kDa. Zymogram analysis of the purified enzyme shows cellulase and xylanase activities, using CMC or xylan as respective substrates, suggesting that TtCel7A is a bifunctional enzyme, manifesting both activities. Furthermore, substrate affinity assays of purified TtCel7A with CMC, xylan, laminarin, or Avicel showed the enzyme to be fully active on CMC, and it also displayed activity on beechwood xylan. However, no activity was detected on laminarin or Avicel. The bifunctional cellulase/xylanase nature of the purified TtCel7A from *T. terrestris* was assessed in terms of the biochemical characterization of both cellulase and xylanase activities.

TtCel7A exhibited optimal activity at pH 5.5, although the enzyme was active in a range of pH from 4.5 to 7 for both cellulase and xylanase activity. Concurring with our data, several fungal cellulases of the GH7 family with optimal acid pH (3.0–6.0) have been reported [4,22,25,45,46,47], and fungal xylanases [19,48] usually have a slightly acid optimal pH (3.0–6.0).

The cellulase activity of TtCel7A displayed better pH stability than that observed for xylanase activity. Considering pH stability, TtCel7A retained more than 80% of cellulase activity at pH 4.5–6.0 after 60 min of incubation, whereas the enzyme retained more than 80% of xylanase activity after only 30 min of incubation at pH 5.0–6.0, and nonxylanase activity was detected for these assays after 60 min of incubation under the same conditions. It is worth mentioning that cellulase activity of TtCel7A from *T. terrestris* Co3Bag1(this work) manifested higher pH stability than that previously reported for the bifunctional enzyme endoglucanase Cel7B from *Trichoderma reesei*, which retained 80% of activity in a pH range of 5.0–6.0, after 30 min of incubation [45]. Xylanase from *Paecilomyces themophila* retained more than 90% of activity after 30 min incubation in a pH range from 6.0 to 11 [49]. In contrast, xylanase from *Chaetomium* sp. retained more than 80% of xylanolytic activity after 30 min of incubation in a pH range from 4.5 to 11 [50].

TtCel7A displayed an optimal cellulolytic temperature of 60 °C and optimal xylanolytic temperature of 50 °C. In agreement with our data, a bifunctional recombinant xylanase/endoglucanase from yak rumen microorganisms from the GH5 family showed different optimum temperatures: 50 °C for endoglucanase activity and 65 °C for xylanase activity [3]. In contrast, bifunctional GH7 family recombinant enzymes, with cellobiohydrolase and xylanase activities, from *Chaetomium thermophilum* display the same optimum temperatures: 55 °C [28] and 60 °C [29] for both enzyme activities and have been described. The property of the same optimum temperature has been explained, as it relates to a unique catalytic domain or a single substrate-binding tunnel [28,29]; however, it has also been mentioned that the same catalytic center does not necessarily guarantee similar pH and temperature ranges [3].

The xylanolytic activity of TtCel7A from *T. terrestris* Co3Bag1 (this work) manifested better thermostability than that observed for cellulolytic activity. TtCel7A retained 50% of xylanolytic activity after 20, 10.3, and 1.1 h of incubation at 50, 60, and 70 °C, respectively, whereas the enzyme retained 50% of cellulolytic activity after 2.1, 1.6, and 0.6 h of incubation at 50, 60, and 70 °C, respectively. The cellulase and xylanase activities of the native TtCel7A from *T. terrestris* (this work) manifested similar thermostability at 60 and 70 °C (retained 83% of activity after 180 min) as that reported for the recombinant thermostable GH7 endoglucanase from *Chaetomium thermophilum* [28]. The cellulase activity of TtCel7A displayed lower thermostability at 60 and 70 °C, respectively, than recombinant endoglucanase GH7 from the fungi *Neosartorya fischeri* [51], which showed 100% of activity after 60 min of incubation at 60 °C, and 16.1% of activity after 60 min of incubation at 70 °C. However, the xylanase activity of the native TtCel7A from *T. terrestris* Co3Bag1 (this work) manifested similar thermostability at 60 °C (t_1/2_ of 10.3 h and retained 83% of activity after 180 min) compared to that reported for the recombinant bifunctional cellobiohydrolase–xylanse from *Chaetomium thermophilum*, which retained more than 80% of its activity after 180 min of incubation at 60 °C [29]. Half-lives of 5 and 9.96 h at 50–60 and 70 °C have been reported for the recombinant GH7 endoglucanase from *Myceliophthora thermophila* [26].

TtCel7A from *T. terrestris* Co3Bag1 can be considered as a bifunctional cellulase–xylanase enzyme due to the enzyme manifested cellulose (33.13 U/mg) and xylanase (1.96 U/mg) activities on CMC and beechwood xylan as the substrates; thus, indicating affinity towards substrates with β-1,4 linkages. Nonactivity was observed on Avicel, microcrystalline cellulose with β-1,4 linkages, as well as laminarin, which contains β-1,3 and β-1,6 linkages. The specific activity values of 26 U/mg and 12 U/mg for cellulase and xylanase activities reported for the endoglucanase I (Cel7B) from *Trichoderma harzianum* [27] are 1.27-fold lower and 6.1-fold higher compared to those described here for cellulase and xylanase activities on CMC and beechwood xylan as the substrates, for TtCel7A from *T. terrestris* Co3Bag1 (this work). In contrast, TtCel7A from *T. terrestris* Co3Bag1 displayed 17.1-fold and 1.3-fold greater cellulase and xylanase activities on CMC and beechwood xylan, respectively, compared to those reported for the GH7 cellobiohydrolase from *Chaetomium thermophilum*, with specific activity values for cellulase and xylanase of 1.94 and 1.49 U/mg, respectively [29].

The kinetic parameters *K*_M_ and V_max_ of TtCel7A from *T. terrestris* Co3Bag1 were determined for cellulase and xylanase activities. The *K*_M_ values for this enzyme were 3.12 and 0.17 mg/mL for cellulase and xylanase activities, respectively, whereas the V_max_ values were 50 and 42.75 U/mg for cellulase and xylanase activities. The data here suggest that TtCel7A manifests an 18-fold higher affinity to xylan compared to CMC, whereas the V_max_ value for CMC was 1.16-fold higher compared to beechwood xylan. For TtCel7A, the catalytic efficiency (*k*_cat_/*K*_M_) was similar for CMC (0.021 mg/mL) and xylan (0.023 mg/mL). However, the turnover rate (*k*_cat_) was 17.9, higher for CMC compared to xylan, indicating that cellulase activity has superior catalytic hydrolysis on CMC as the substrate. Our data concur with those reported for the bifunctional cellobiohydrolase–xylanase (CtCel7) GH7 from *Chaetomium thermophilum* [29], where the *k*_cat_ on glucan (*k*_cat_ = 1.54 ± 0.29 (s^−1^)) was 1.7-fold greater compared to xylan (*k*_cat_ on xylan was 0.90 ± 0.05 (s^−1^)). 

The differential spectrum for CMC suggests some changes in the secondary structure of TtCel7A in the presence of CMC as the substrate; in contrast, the differential spectrum for beechwood xylan indicates there was no major secondary structure reorganization for TtCel7A when beechwood xylan was the substrate. The changes in the secondary structure of TtCel7A in the presence of CMC as the substrate may be related to the higher substrate specificity and greater turnover rate (*k*_cat_) of the enzyme with CMC as the substrate compared to xylan.

The effect of metal ions on the cellulase and xylanase activities of TtCel7A from *T. terrestris* Co3Bag1 was analyzed. For cellulase activity, an increase of 126, 83, 78, and 59% was observed in the presence of the metal ions Mn^2+^, Co^2+^, Fe^2+^, and Ca^2+^, respectively, whereas the ion Hg^2+^ completely inhibited this enzymatic activity. Concurring with our findings, an increase of 103.2 and 37.2% in the cellulase activity of the recombinant thermostable GH7 endoglucanase from *Chaetomium thermophilum* was observed in the presence of the ions Mn^2+^ and Co^2+^ [28]. Likewise, the fact that the metal ion Hg^2+^ is toxic to enzymes has been described, as it binds to thiol groups present in the active sites of the enzyme, causing irreversible inactivation [52]. For xylanase activity, an increase of 44, 16, and 15% was observed with the ions Fe^2+^, Mn^2+^, and K^1+^, respectively. Likewise, an increase in the xylanase activity (47.5%) was reported in the presence of Mn^2+^ (1 mM) [28]. An inhibitory effect on the glycoside hydrolases activity has been associated with the metal ion Cu^2+^, particularly for the formation of inter- and intramolecular disulfide bridges, leading to oxidation of cysteine residues; however, there is no pattern to describe the effect of metal ions on cellulase and xylanase activities [29].

Hydrolysis products on CMC, xylan, and complex substrates were analyzed to determine the exo- or endocatalytic mode of action of TtCel7A. The main hydrolysis product of CMC released by TtCel7A was glucose, and a small amount of high molecular weight (MW) oligosaccharides was also observed. Xylobiose was thought to be the main product yielded; however, a small amount of xylose and xylo-oligosaccharides was also released when beechwood xylan was used as the substrate.

Further studies were carried out using agroindustrial derivatives such as oat bran, wheat bran, and sugarcane bagasse. Interestingly, the main hydrolysis products from oat and wheat bran were glucose and a small number of xylo-oligosaccharides. In contrast, when sugarcane bagasse was the substrate, glucose and cellobiose were detected as the main products yielded; also, a small amount of xylose and xylo-oligosaccharides were detected. Hence, our results suggest TtCel7A displayed marked exocellulase and exoxylanase activity, even though slightly less endocellulase and -xylanase activities were apparent; thus, confirming the bifunctional properties of the enzyme. Likewise, there are some reports of bifunctional enzymes such as the bifunctional recombinant cellulase–xylanase (rBhcell-xyl) from the bacterium *Bacillus halodurans*, where xylose, xylobiose, xylotriose, xylotetraose, and xylopentaose were described as the main products released when xylan was the substrate, as well as cellobiose from CMC, thus, suggesting an endoxylanase–endocellulase and exocellulase activity on the part of the recombinant enzyme [10]. It has been reported that recombinant GH7 endoglucanase from *Chaetomium thermophilum* showed that the main products liberated on CMC and xylan were monosaccharides, disaccharides, and trisaccharides, suggesting an endoaction mode with the CTendo7 enzyme [28].

Concurring with our results, phylogenetic analysis of the bifunctional cellulase–xylanase Cel7A from *T. terrestris* Co3Bag1 and other glycoside hydrolases from GH3, GH5, and GH7 allowed us to identify the enzyme as a member of the GH7 family.

Additionally, the bifunctional cellulase–xylanase TtCel7A from *T. terrestris* Co3Bag1 enzyme showed modular structure, characteristic of the cellulases, such as peptide signal, catalytic domain, cellulose binding domain, and linker region, a conserved modular structure that has been reported for other glycoside hydrolases of the GH7 family [13]. According to the secondary structure analysis, the putative N- and O-glycosylation sites mainly take place on the region that links CD and CBD of the enzyme. This region was rich in Ser and Thr amino acid residues, concurring with previous reports [6,26,29,53].

The CD structure consists of a β-sandwich with antiparallel β-sheets with a typical tunnel-like catalytic domain similar to the β-jelly roll observed for glycoside hydrolases of the GH7 family reported for fungi [22,29,46].

Bifunctional cellulases–xylanases have been reported in GH7 enzymes [26,27,28,29]. There are a few reports of cellulases from *T. terrestris* [33,54], but to the best of our knowledge, no studies are related to a native bifunctional cellulase/xylanase GH7 family from members belonging to the genus *Thielavia* manifesting both cellulase and xylanase activities.

According to the characteristics reported in this work for the bifunctional cellulase-xylanase TtCel7A from *T. terrestris* Co3Bag1 with exocellulase, exoxylanase, and slight endoxylanase catalytic mode of action, the enzyme might be considered a good candidate for the hydrolysis of lignocellulosic material, as the enzyme displayed both thermophilic cellulolytic and xylanolytic activities, with thermostability and activity in a wide pH range for both enzymatic activities. Furthermore, we include experimental evidence to demonstrate the capability of the enzyme TtCel7A for the hydrolysis of complex substrates, such as oat bran, wheat bran, and sugarcane bagasse, releasing glucose, cellobiose, and xylose as the main products. Moreover, this work represents the first report describing a bifunctional cellulase–xylanase TtCel7A from *T. terrestris* Co3Bag1.

## Figures and Tables

**Figure 1 jof-09-00152-f001:**
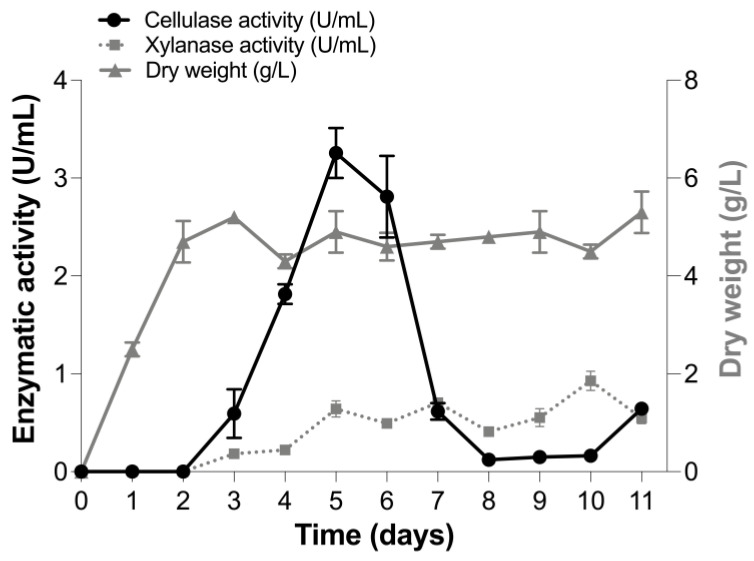
Cellulase and xylanase activity and dry weight of *T. terrestris* Co3Bag1 at 45 °C. Cellulase activity (black circle), xylanase activity (gray square), and dry weight (gray triangle) were produced on a Zouari-Mechichi medium with 1% (*w*/*v*) CMC as a carbon source.

**Figure 2 jof-09-00152-f002:**
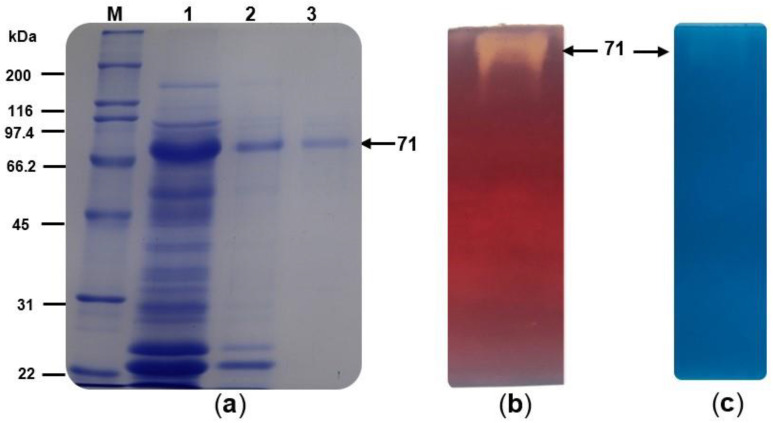
Purification and zymogram analysis of TtCel7A on 10% SDS-PAGE. (**a**) A 10% SDS-PAGE analysis of purified TtCel7A. Lanes M: molecular weight standard; 1: crude extract; 2: anion exchange chromatography fractions; 3: purified TtCel7A. (**b**) Zymogram analysis of TtCel7A using 1% CMC as substrate. (**c**) Zymogram analysis of TtCel7A, using 1% RBB-X as substrate.

**Figure 3 jof-09-00152-f003:**
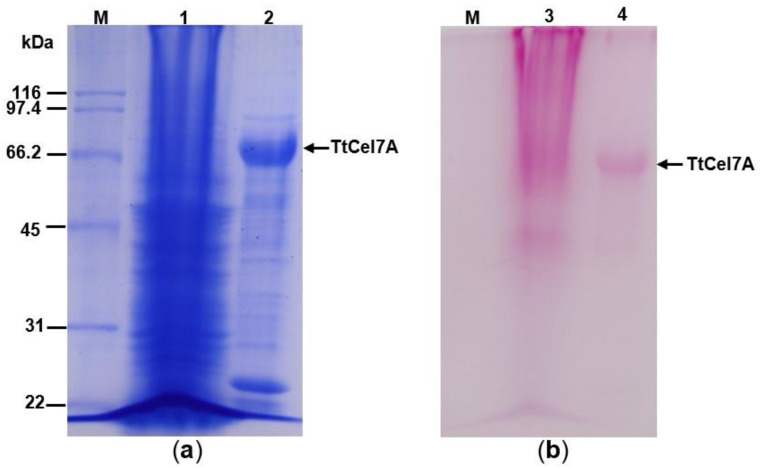
(**a**) A 10% SDS-PAGE analysis of semipurified TtCel7A. (**b**) Glycoprotein analysis, Schiff stained of semipurified TtCel7A. Lanes M: molecular weight standard; 1 and 3: Invertase from *Saccharomyces cerevisiae* as control; 2 and 4: semipurified TtCel7A.

**Figure 4 jof-09-00152-f004:**
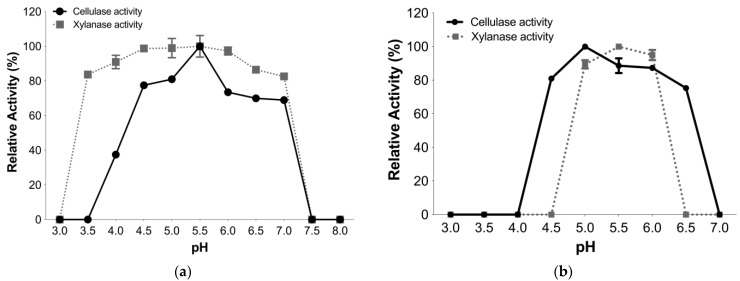
Influence of pH and temperature of TtCel7A. (**a**) pH effect using different buffers: citrate (pH 3–6), acetate (pH 4–5.5), citrate-phosphate (pH 3–7), and phosphate (pH 5.5–8) incubated at 60 °C and 10 min for cellulase activity and 50 °C and 15 min for xylanase activity. (**b**) Influence of pH stability on TtCel7A incubated in citrate-phosphate buffer (pH 3–7) at 25 °C for 60 min for cellulase and 30 min for xylanase activity.

**Figure 5 jof-09-00152-f005:**
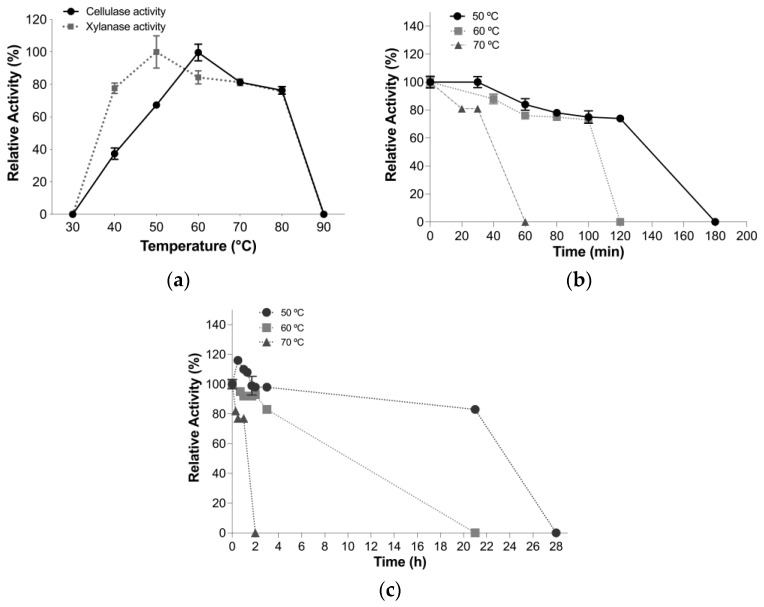
(**a**) Temperature profile of cellulase and xylanase activity of TtCel7A, incubated with 0.6% (*w*/*v*) of the substrate in 50 mM acetate buffer, pH 5.5. (**b**) Thermal stability of cellulase activity of TtCel7A incubated at 50 (circle), 60 (square), and 70 °C (triangle) without substrate. (**c**) Thermal stability of xylanase activity of TtCel7A incubated at 50 (circle), 60 (square), and 70 °C (triangle) without substrate.

**Figure 6 jof-09-00152-f006:**
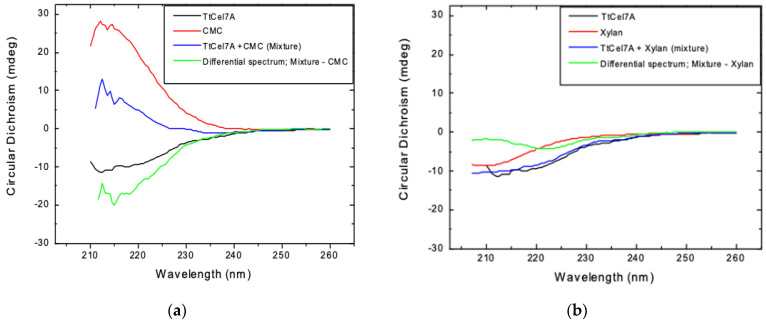
(**a**) Far-UV CD spectra of the purified TtCel7A, CMC, or TtCel7A+CMC mixture and (**b**) TtCel7A, beechwood xylan, or TtCel7A+ beechwood xylan mixture were measured using a JASCO J-815 spectropolarimeter (Jasco Inc., Easton, MD) equipped with a PFD-425S Peltier-type cell holder for temperature control and magnetic stirring. Scans were taken between 200 and 250 nm, at a scan rate of 10 nm min^−1^ using a 1.0 cm path-length cuvette. Buffer baselines were subtracted from all spectra.

**Figure 7 jof-09-00152-f007:**
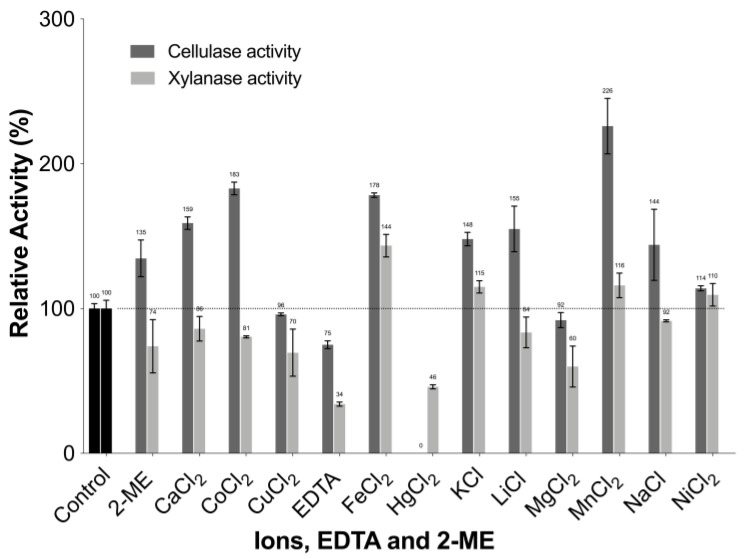
Effect of metal ions, EDTA, and 2-ME on cellulase and xylanase activity of TtCel7A. Activity was measured at 1 mM concentration and incubated in 0.6% (*w*/*v*) CMC and xylan in 50 mM acetate buffer, pH 5.5. The activity observed without any element was used as a control and considered 100%.

**Figure 8 jof-09-00152-f008:**
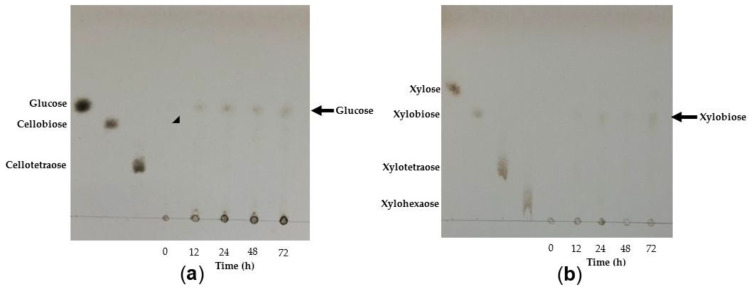
TLC analysis of the hydrolysis products released by the action of TtCel7A at 40 °C from (**a**) CMC, (**b**) beechwood xylan at 1% concentration after 12 to 72 h incubation.

**Figure 9 jof-09-00152-f009:**
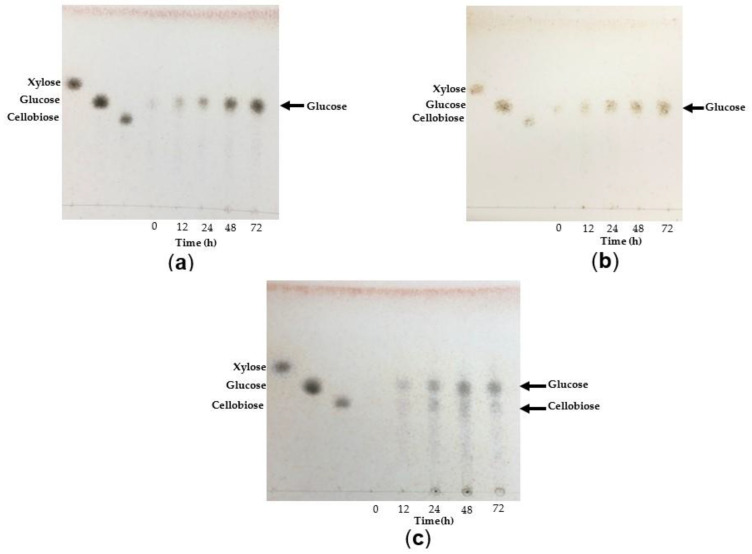
TLC analysis of the hydrolysis products released by the action of TtCel7A at 40 °C from (**a**) oat bran, (**b**) wheat bran, and (**c**) sugarcane bagasse at a final concentration of 30 mg/mL.

**Figure 10 jof-09-00152-f010:**
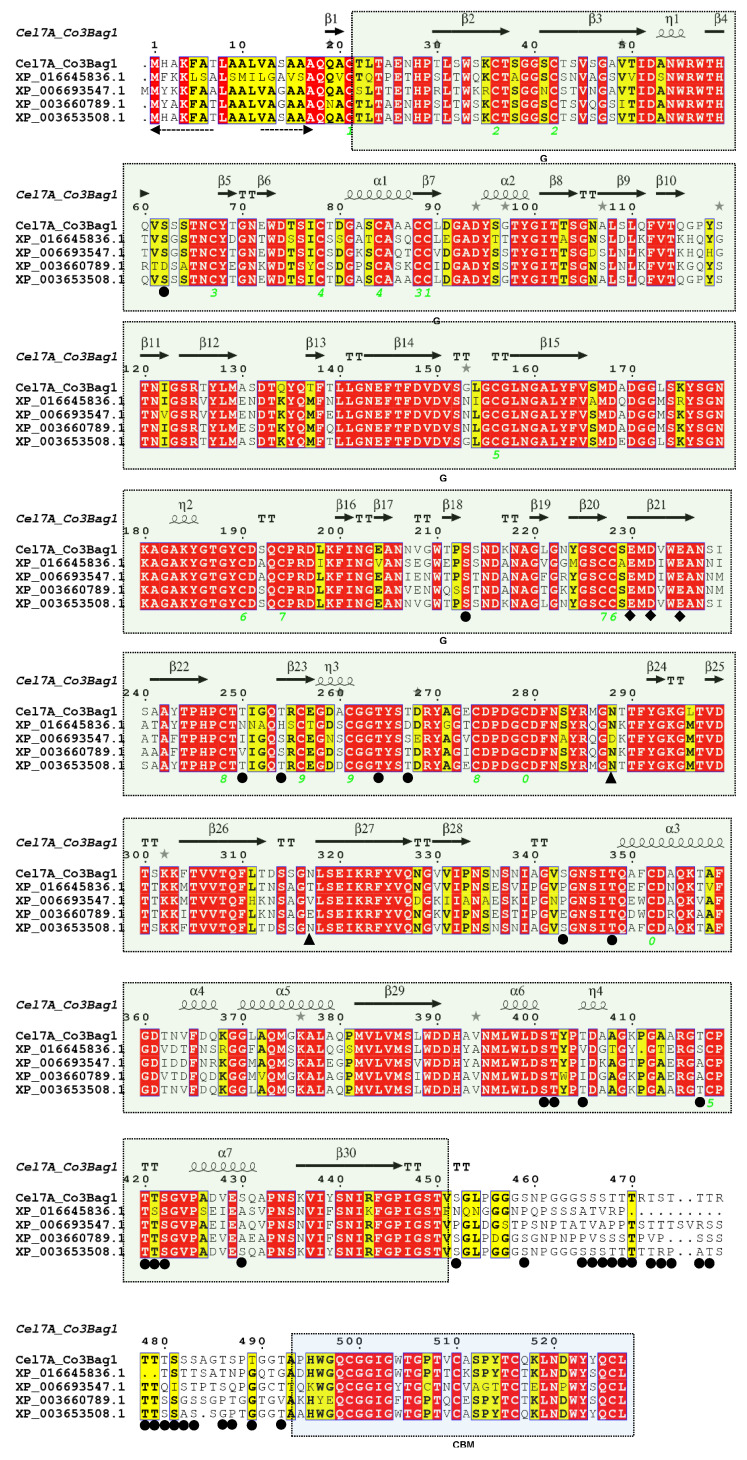
Multiple sequence alignment and secondary structure of TtCel7A and other enzymes. Sequence alignment of two GH7 enzymes from *Thermothielavioides terrestris* NRRL 8126 (XP_003653508.1) and *Thermothelomyces thermophilus* ATCC 42464 (XP_003660789.1) and two exoglucanases from *Scedosporium apiospermum* (XP_016645836.1) and *Chaetomium thermophilum* var. *thermophilum* DSM 1495 (XP_006693547.1). For the secondary structure of TtCel7A, β-strands and α-helices were based on the 3D crystal structure of the thermostable *cellobiohydrolase* Cel7A from the fungus *Humicola grisea* var. *thermoidea*. Strictly conserved residues are seen in the red box and similar residues in the yellow box. The catalytical residues Glu230, Asp232, and Glu235 are represented with diamonds. The triangle and circles indicate the probable N- and O-glycosylated sites. The sequence alignment was generated with the ESPript program.

**Figure 11 jof-09-00152-f011:**
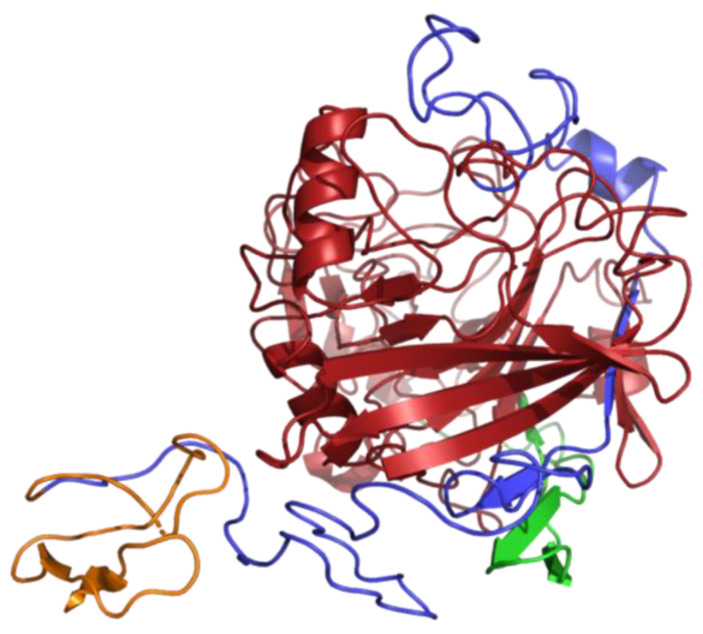
The 3D predicted structure of TtCel7A from *T. terrestris* Co3Bag1, obtained with the Phyre^2^ server. The CD was identified with red, the CBM with orange, and the linker region with blue color.

**Figure 12 jof-09-00152-f012:**
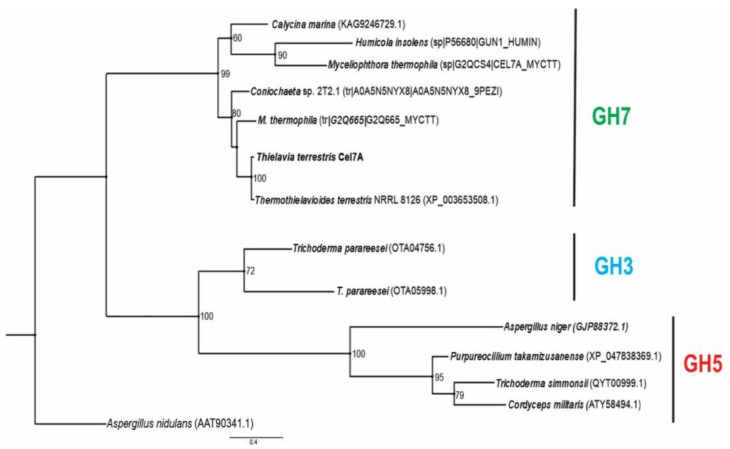
Maximum likelihood phylogenetic tree of the bifunctional cellulase/xylanase TtCel7A amino acid sequence from *T. terrestris* Co3Bag1 and the representative sequences of GenBank and Uniprot databases. The Blosum62+G+F model (−InL 14,779.55, gamma parameter 8.079) was used for the analysis. As an outgroup, the putative endoglucanase sequence of *A. nidulans* of the GH17 family (AAT90341.1) was included. The robustness at each node was assessed after 1000 pseudoreplicates, and bootstrap support values are indicated for major nodes having 50% values. The scale bar indicates substitution/site.

**Table 1 jof-09-00152-t001:** Purification steps of TtCel7A from *T. terrestris* Co3Bag1.

Purification Step	Total Activity (U)	Total Protein (mg)	Specific Activity (U/mg)	Yield (%)	Purification Fold
Crude extract	124	239	0.52	100	1.0
80% (NH4)_2_SO_4_ precipitation	26.00	45.61	0.57	20.96	1.10
Anion exchange chromatography	10.85	1.73	6.27	8.75	12.06
Gel filtration chromatography	7.62	0.23	33.13	6.15	63.71

**Table 2 jof-09-00152-t002:** Kinetic properties of TtCel7A from *T. terrestris* Co3Bag1.

Substrates	*K*_M_ (mg/mL)	V_max_ (U/mg)	Catalytic Efficiency (mL/mg s)	*k_cat_* (s^−1^)
CMC ^a^	3.12	50	0.021	0.07
Beechwood xylan ^b^	0.17	42.75	0.023	3.9 × 10^−3^

^a^ on acetate buffer, pH 5.5, containing 0.6% (*w*/*v*) of CMC, incubated at 60 °C for 10 min; ^b^ on acetate buffer, pH 5.5, containing 0.3% (*w*/*v*) of beechwood xylan, incubated at 50 °C for 15 min.

## Data Availability

Not applicable.

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
