# Peer review of "TtCel7A: A Native Thermophilic Bifunctional Cellulose/Xylanase Exogluclanase from the Thermophilic Biomass-Degrading Fungus Thielavia terrestris Co3Bag1, and Its Application in Enzymatic Hydrolysis of Agroindustrial Derivatives"

_jof, 2023, doi:10.3390/jof9020152_

Round 1

Reviewer 1 Report

López-López et al., authors of a submitted manuscript, “titled: TtCel7A: a native thermophilic bifunctional cellulose/xylanase exogluclanase from the thermophilic biomass-degrading fungus Thielavia terrestris Co3Bag1, and its application in enzymatic hydrolysis of agroindustrial derivatives” studied a native GH7 bifunctional cellulase-xylanase from T. terrestris Co3Bag1 and the application of the enzyme in the enzymatic hydrolysis of agroindustrial derivatives, such as oat bran, wheat bran, and sugarcane bagasse.

A submitted manuscript was good in experiments design, procedure, and conclusion. Nonetheless, it needs some minor revisions.

1-    Please add all primes used in this study in the supplementary file.

2-    Please add the details of genomic DNA extraction used in this study.

3-    Also, describe the details of how to get the TtCel7A sequence using PCR.  

Author Response

Reviewer 1

Comments and Suggestions for Authors

López-López et al., authors of a submitted manuscript, "titled: TtCel7A: a native thermophilic bifunctional cellulose/xylanase exogluclanase from the thermophilic biomass-degrading fungus Thielavia terrestris Co3Bag1, and its application in enzymatic hydrolysis of agroindustrial derivatives" studied a native GH7 bifunctional cellulase-xylanase from T. terrestris Co3Bag1 and the application of the enzyme in the enzymatic hydrolysis of agroindustrial derivatives, such as oat bran, wheat bran, and sugarcane bagasse.

A submitted manuscript was good in experiments design, procedure, and conclusion. Nonetheless, it needs some minor revisions.

Answer to Reviewer 1 comments.

1-    Please add all primes used in this study in the supplementary file.

<Done. Following your suggestion, all primers employed for DNA sequencing of the Ttcel7A gene were included in Table S1 in the supplementary file.>

2-    Please add the details of genomic DNA extraction used in this study.

<Done. We apologize for any confusion we may cause because the information provided in Sections 2.9 and 2.10 (manuscript jof-2142636) was unclear. Sections 2.9 and 2,10 were rewritten for improved clarity in the revised version of this manuscript jof-2142636 - Minor Revisions.

Now, in the revised version of the manuscript jof-2142636 (Sections 2.9, 2.10 and 2.11) we present the details of how to obtain the DNA sequence of the TtCel7A encoding gene by reverse genetics, using cDNA from T. terrestris Co3Bag1 as the template for PCR amplification. The DNA sequence of the TtCel7A encoding gene was confirmed by bioinformatic analysis of the genomic sequence from T. terrestris Co3Bag1.

Note: The following information that was not included in the revised version of the manuscript jof-2142636.

For extraction of genomic DNA, T. terrestris Co3Ba1 was grown at 45 °C for 48 h and 200 rpm in 50 mL of medium containing malt extract (10 g/L), yeast extract (4 g/L) and glucose (4 g/L) in a 250 ml Erlenmeyer flask. Biomass was obtained by centrifugation at 13 000 rpm for 5 min and resuspended in 500 mL of sterile deionized water. The isolation of genomic DNA was carried out by using ZR Soil Microbe DNA Kit and following the manufacturer's instructions. Genomic DNA was analyzed in 1 % agarose gel and visualized under ultraviolet light after staining with ethidium bromide. Gel image acquisition was performed using Doc-it SL Image Acquisition Software. The whole genome sequencing was performed on the HiSeq2000 Illumina platform by Otogenetics Corporation (Norcross, GA, United States) (unpublished results). Currently, we are working on the heterologous expression of the TtCel7A encoding gene from T. terrestris Co3Bag1.>

3-    Also, describe the details of how to get the TtCel7A sequence using PCR.

<Done. The conditions for the amplification of the coding region of the TtCelA encoding gene were included in section 2.10 of the revised version of the manuscript.>

Reviewer 2 Report

This article investigates a bifunctional cellulase/xylanase from the GH7 family produced by the thermophile Thielavia terrestris. A comprehensive study is carried out, including enzyme production profiles, enzyme purification, optimal pH and temperature, stability in differente pH and temperature conditions, substrate specificity, kinetics, gene analysis, partial aminoacid sequencing, 3D modelling and phylogenetic reconstruction.

The information provided could be useful for further development of biorefinery processes. The method are adequately explained, and the discussion is sound. In summary, this manuscript could be published in JoF, provided that some minor flaws are corrected:

- English language needs in-depth revision; there are a number of errors that should be corrected (particularly in verbs and articles)

- Please use the same symbol for degree C throughout the manuscript (e.g. see line 70)

- A space should be written between the number and the symbol % (this is not always the case in the manuscript).

- Some units are incorrect (see line 113, the "micro" symbol is missing

- Line 130: proteins are adsorbed, not absorbed

- In the purification chapter, chromatograms should be provided (I suggest including them in the Supplementary material document)

- The quality of all figures should be revised and improved (the scales and the titles of axis are frequently superposed and/or difficult to read

Author Response

Journal: Journal of Fungi

Manuscript ID: jof-2142636 - Minor Revisions

Title: TtCel7A: a   thermophilic bifunctional cellulose/xylanase exogluclanase from the thermophilic biomass-degrading fungus Thielavia terrestris Co3Bag1 and its application in enzymatic hydrolysis of agroindustrial derivatives.

Dear Editor,

Thank you very much for the opportunity to prepare the revised version of the manuscript ID: jof-2142636 - Minor Revisions. Also, we appreciate the time of Reviewer 1 and Reviewer 2 to review the manuscript as mentioned above; their comments and suggestions complete and improve the quality of the manuscript.

It is worth mentioning that we include Yolanda García-Huante as co-author of this work due she carried out the sequencing of the TtCel7A encoding gene (Ttcel7A) during her Ph.D. studies under my supervision. The nucleotide sequence of the Ttcel7A was reported in GenBank in 2019 (accession number: KX426377.1)>

I will be looking forward to hearing from you.

Best regards,

María E. Hidalgo-Lara

Reviewer 2

Comments and Suggestions for Authors

This article investigates a bifunctional cellulase/xylanase from the GH7 family produced by the thermophile Thielavia terrestris. A comprehensive study is carried out, including enzyme production profiles, enzyme purification, optimal pH and temperature, stability in different pH and temperature conditions, substrate specificity, kinetics, gene analysis, partial amino acid sequencing, 3D modeling, and phylogenetic reconstruction.

The information provided could be useful for further development of biorefinery processes. The method are adequately explained, and the discussion is sound. In summary, this manuscript could be published in JoF, provided that some minor flaws are corrected:

Answer to Reviewer 2 comments.

- English language needs in-depth revision; there are a number of errors that should be corrected (particularly in verbs and articles)

<Done. Following your suggestion, the manuscript has been edited by an English-speaking native, so we hope the grammar mistakes are now solved in the revised version of the manuscript jof-2142636.>

- Please use the same symbol for degree C throughout the manuscript (e.g. see line 70)

<Done. Thank you for your observation. The same symbol for the degree (ºC) was used throughout the revised version of the manuscript.>

- A space should be written between the number and the symbol % (this is not always the case in the manuscript).

<Done. Following your suggestion, now there is a space written between the number and the symbol % throughout the revised version of the manuscript>

- Some units are incorrect (see line 113, the "micro" symbol is missing

<Done. Thank you very much for your observation. The missing symbol (µ) was written in section 2.3, and the complete manuscript was revised to correct any other missing symbol.>

- Line 130: proteins are adsorbed, not absorbed

<Done. Thank you for your observation. The word "absorbed" was changed to "adsorbed" as suggested.>

- In the purification chapter, chromatograms should be provided (I suggest including them in the Supplementary material document)

<Done. Following your suggestion, the chromatograms obtained during the purification of Cel7A by anion exchange chromatography and gel filtration chromatography were included as Supplementary material (Fig. S4) in the revised version of the manuscript.>

- The quality of all figures should be revised and improved (the scales and the titles of the axis are frequently superposed and/or difficult to read

<Done. Following your suggestion, all figures were revised and modified accordingly for improved clarity concerning the scales and the titles of the axis, making the figures easier to read. Also, the resolution of the figures was improved for better quality.>